# Double Perovskite LaFe_1−x_Ni_x_O_3_ Coated with Sea Urchin-like Gold Nanoparticles Using Electrophoresis as the Photoelectrochemical Electrode to Enhance H_2_ Production via Surface Plasmon Resonance Effect

**DOI:** 10.3390/nano12040622

**Published:** 2022-02-12

**Authors:** Hsiang-Wei Tsai, Yen-Hsun Su

**Affiliations:** Department of Materials Science and Engineering, National Cheng-Kung University, Tainan City 701401, Taiwan; n58064026@gs.ncku.edu.tw

**Keywords:** LaFe_1−x_Ni_x_O_3_, gold nanoparticles, electrophoresis, hydrogen production, photoelectrochemical

## Abstract

The surface plasmon resonance (SPR) effect and the hetero-junction structure play crucial roles in enhancing the photocatalytic performances of catalysts for the water-splitting reaction. In this study, a series of double perovskites LaFe_1−x_N_ix_O_3_ was synthesized. LaFe_1−x_N_ix_O_3_ particles were then decorated with sea urchin-like Au nanoparticles (NPs) with the average size of approximately 109.83 ± 8.48 nm via electrophoresis. The d-spacing became narrow and the absorption spectra occurred the redshift phenomenon more when doping increasing Ni mole concentrations for the raw LaFe_1−x_N_ix_O_3_ samples. From XPS analysis, the Ni atoms were inserted into the lattice of the matrix, resulting in the defect of the oxygen vacancy, and NiO and Fe_2_O_3_ were formed. This hybrid structure was the ideal electrode for photoelectrochemical hydrogen production. The photonic extinction of the Au-coated LaFe_1−x_Ni_x_O_3_ was less than 2.1 eV (narrow band gap), and the particles absorbed more light in the visible region. According to the Mott–Schottky plots, all the LaFe_1−x_N_ix_O_3_ samples were the n-type semiconductors. Moreover, all the band gaps of the Au-coated LaFe_1−x_N_ix_O_3_ samples were higher than 1.23 eV (H^+^/H_2_). Then, the hot electrons from the Au NPs were injected via the SPR effect, the coupling effect between LaFe_1−x_N_ix_O_3_ and Au NPs, and the more active sites from Au NPs into the conduction band of the semiconductor, improving the hydrogen efficiency. The H_2_ efficiency of the Au-coated LaFe_1−x_Ni_x_O_3_ measured in ethanol was approximately ten times larger than the that of Au-coated LaFe_1−x_Ni_x_O_3_ measured in 1-butanol at any testing temperature because ohmic and kinetic losses occurred in the latter solvent. Thus, the activation energies of ethanol at any testing temperature were smaller. The maximum real H_2_ production was up to 43,800 μmol g^−1^ h^−1^ in ethanol. The redox reactions among metal ions, OH*, and oxides were consecutively proceeded under visible light illumination.

## 1. Introduction

The development of alternative catalysts for the oxygen evolution reaction (OER) and hydrogen evolution reaction (HER) by tuning the structure and composition of catalysts based on transition metals, for example, metal oxides (or hydroxides), phosphates, chalcogenides, and metal–carbon materials [1], has attracted significant research attention. In particular, materials with the ABO_3_ perovskite structure have been found to serve as catalysts with high activities, selectivities, and stabilities [2] owing to their optimum electronic and chemical structures. Especially, double perovskites have received attention in recent years, because double perovskites (AA’BB’O6), in which the A- and B-site elements are substituted by a transition metal, possess several intriguing chemical and physical properties, such as electronic structures that range from insulating to metallic, half-metallic spin-polarized electrical conductivity, and superconductivity [3]. Guan et al. also reported that B-sites coordinated with O anions are considered as the p-band center, and are also identified as the active sites for absorption and desorption reaction intermediates [4]. Accordingly, the double perovskite structure of LaFe_1__−__x_Ni_x_O_3_ has been extensively studied. In this structure, Ni^3+^ is in the low spin state (t^6^_2g_, e^1^_g_), and the conduction band is formed by the hybridization of Ni^3+^ e_g_ and O^2^^−^ 2p orbitals [5], which provide the design and selection for the photocatalytic applications. To compare with traditional materials, double perovskites possess the superior performance about the screening effect. The better the screening effect, the lower the probability of charge recombination and the stronger the B-O covalency due to the impurity of Ni. This is because d holes of electrons are in s and p states of Ni, and the impurity of charge is screened mainly by d electrons at surrounding Ni atoms [6]. This research has demonstrated that the HER activity of an Ni surface can be reinforced by NiO, or Ni(OH)_2_ formation due to HO-H bonds, and then H^+^ is profoundly and proficiently converted into H_2_ [7].

The photocatalytic performances of perovskites can be considerably improved by tuning their surface areas and intrinsic activities [8,9]. The electronic properties of perovskites are also important factors [10]. Simultaneous reduction–oxidization reactions are facilitated by tailoring their electronic structure to permit the easy and rapid transport of electrons to the surfaces of the catalysts [11,12,13]. The band gap of the optimal photocathode is approximately 1.2 eV to establish the optimum HER process [14]. This kind of photocathode possesses high solar-to-hydrogen (STH) efficiency and the unassisted water-splitting reaction, which means that there is only one key step of artificial photosynthesis converting solar light energy into chemical fuels [15]. There are some factors, which were the electronic band gap, exposed active sites, and diverse and adjustable chemical structure, to improve the efficiency of the photoelectrochemical (PEC) cell [16].

Many methods can be used to develop the efficiency of the water-splitting reaction. The optical property of metal nanoparticles which induce the SPR effect is generally taken advantage of. The sensitivity of SPR, as well as its spatial distribution and resonant energy, depend on the nanoparticle structure, composition, and environment [17]. The light–matter interaction occurs when Au NPs are incorporated into the photoactive material at the interface between the semiconductor and Au NP The enhanced scattering and absorption efficiencies result from the excitation of SPR with the stronger near-field intensity at the boundary of Au nanostructures [18]. The sharp, well-defined tips and spikes in star-shaped nanoparticles (NPs) result in enhanced surface Raman scattering, and consequently, these structures exhibit improved SPR effect [19].

This study compared the activation energy of LaFe_1−x_Ni_x_O_3_ in different solvents. LaFeO_3_ were doped with varying concentrations of Ni to synthesize the double perovskite structure of LaFe_1−x_Ni_x_O_3_ with a narrow band gap. Electrophoresis was used to synthesize urchin-like Au NPs, and a hetero-junction photocathode was used to combine LaFe_1−x_Ni_x_O_3_ with Au NPs. To use the electrophoresis method was convenient and non-expensive. This hybrid structure demonstrated appropriate stability for water splitting. Such materials have been used to catalyze reactions involving short- and long-chain hydrocarbons, ethanol and 1-butanol, respectively, to generate the photocatalytic H_2_ production. The results obtained herein can aid in enhancing the application of perovskites in the energy conservation field. The primary aims of this study were (1) 3D bulk structure of LaFe_1−x_NixO_3_ synthesis using the hydrothermal method; (2) LaFe_1−x_NixO_3_ decorated with Au nanoparticles, forming the hetero-junction electrode; (3) light-driven hydrogen generation in ethanol and 1-butanol tested; (4) the charges of Fe, Ni, and hydroxyl transferring each other; and (5) SPR effect of Au nanoparticles having the key role in catalytic behavior.

## 2. Materials and Methods

### 2.1. Synthesis of LaFe_1−x_Ni_x_O_3_ (LFNO) Bulk Materials

First, 0.2 mol La(NO_3_)_3_•6H_2_O (Alfa Aesar, Tewsbury, MA, USA, 99.9%) and 0.2 mol Fe(NO_3_)_3_•9H_2_O (Alfa Aesar, Tewsbury, MA, USA, 98.0%) were mixed with 160 mL of deionized water (DI water, Thermo-Scientific, Rochester, NY, USA, MicroPure UV, 18.2 MΩ-cm). Additionally, Ni(NO_3_)_2_•6H_2_O (Alfa Aesar, Tewsbury, MA, USA, 98%) was added. LaFe_1−x_Ni_x_O_3_ materials with La:Fe:Ni in the molar ratio of 1:1−x:x, where x = 0.00, 0.01, 0.03, 0.05, and 0.07, were prepared. Subsequently, the solution was stirred for 30 min at 25 °C, after which 0.5 g of polyvinylpyrrolidone (PVP, Sigma-Alorich, St. Louis, MO, USA, 99%) was added, and the mixture was stirred for 1 h at 25 °C. The resulting solution was poured into an autoclave, which was then placed in a drying oven (WahFu, Taiwan, JB-27) at 160 °C for 12 h. After the hydrothermal reaction, the obtained powders, which were directly reacted in the mixture, were washed using DI water and alcohol five times in a centrifuge (HsiangTai, Taiwan, CN-820) at 4000 rpm. Finally, the powders were placed on a baker and dried in the drying oven at 150 °C for 4 h. Using the hydrothermal method can result in high purity, high quality, and narrow particle size. The hydrothermal method can also control stoichiometry, distribution, morphology, uniformity, cause fewer defects, dense particles, high crystallinity, excellent reproducibility, microstructure, high reactivity, and sintering ability. [20]. From Yoshimura’s research [21], the particle size of BaTiO_3_ using the hydrothermal method was the smallest compared with other synthesis processes, such as ball-milling, solid-state reaction, co-precipitation, etc.

### 2.2. Spin-Coating LFNO Powder on ITO (Indium-Tin-Oxide) Glass

First, ITO glass (Ruilong, Taiwan, 7Ω) with the dimensions 20 × 20 × 0.7 mm^3^ was treated using ozone for 10 min. A gel solution with the following components was prepared: 0.1 g of LFNO, 500 μL of Nafion (Aldrich, Milwaukee, WI, USA, 5 wt %), and 3 mL of DI water. The gel solution was stirred for 2 h at 25 °C, after which 45 μL of the gel solution was dropped with a pipe in a spin-coater (Apisc, Taiwan, SP-M1-S) running at 6500 rpm for 15 s at 25 °C. The coated glass was then dried in the oven at 150 °C for 6 h.

### 2.3. Decoration of Sea Urchin-like Au NPs

Electrophoresis was conducted using an electrochemical analyzer (CHI, Model 6000E Series, Austin, TX, USA). Coated glass was used as the working electrode, and a Pt wire was connected with the counter and reference electrodes. An Au colloid was prepared by mixing 50 mL of 5 × 10^−4^ M HAuCl_4_•3H_2_O (Alfa Aesar, Tewsbury, MA, USA, 99.99%) and 5 mL of 4.5 × 10^−2^ M sodium citrate dihydrate (J.T Baker, Suite 100 Bridgewater, NJ, USA, 99.0%). Thereafter, an Au solution consisting of 90 μL of Au colloid, 45 mL of 10^−4^ M HAuCl_4_•3H_2_O, 7.5 mL of 3.0 × 10^−3^ hydroquinone (J.T Baker, Suite 100 Bridgewater, NJ, USA, 99.0%), and 60 μL of 2 wt % sodium citrate dihydrate was prepared. The solution was mixed thoroughly, and the coated glass was introduced into the Au solution. A voltage of 2 V was applied for 2 h for electrophoresis. There were some reasons to use electrophoresis method: (1) electrophoresis is direct and not complicated to load the Au NPs on the thin films, (2) electrophoresis can save time, (3) electrophoresis only needs a small volume of samples, and (4) electrophoresis has a reasonable cost. In this study, the larger surface area, which induces more active sites, and the star-like morphology of Au NPs can enhance the SPR effect.

### 2.4. A Analysis of Properties

The morphology was observed via scanning electron microscopy (SEM, ZEISS-AURIGA AFE-SEM, Jena, Germany); energy dispersive X-ray spectroscopy (EDS) was conducted in conjunction with SEM. The crystal planes were observed via high-resolution transmission electron microscopy (HR-TEM, JEOL JEM-3010, Tokyo, Japan). X-ray photoelectron spectroscopy (XPS, Thermo K-Alpha, X-ray source: Al K-α, Rochester, NY, USA) was conducted to obtain the binding state of each element. The valence band energies of the compounds were obtained from ultraviolet photoelectron spectroscopy (UPS) plots (Thermo VG-Scientific, Sigma Probe, UPS source: He(Ⅰ), Thermo VG-Scientific). The photon energy of the incident light was 21.1 eV.

Ultraviolet-visible (UV-vis) spectroscopy (PerkinElmer, Lambda 950, Waltham, MASS, USA) was conducted to obtain the absorbance wavelengths and band gaps from 400 to 800 nm. The crystal phases were analyzed using an X-ray analytical system (Bruker AXS Gmbh, Karlsruhe, Germany). Electrochemical analysis was conducted using several methods, such as Mott-Schottky plots, cyclic voltammetry (C-V) curves, and linear sweep voltammetry (LSV) curves. The C-V and LSV curves were measured in 0.1 M KOH solution. LSV was conducted under illumination air mass (AM) 1.5G (1000 W/m^2^). Furthermore, the H_2_ evolutions in ethanol (J.T Baker, Suite 100 Bridgewater, NJ, USA, 99.9%) and 1-butanol (Duksan, Seoul, South Korea,99.9%) were determined from the corresponding I–V curves, with the applied voltage ranging from 0 to 1.5 V; testing was also conducted at different temperatures under AM 1.5G.

## 3. Results and Discussion

### 3.1. Phase and Morphology Analyses

Figure 1 shows the XRD analysis of the raw LFNO. From Figure 1 the diffraction peaks can be matched to the orthorhombic perovskite structure with the Pnma space group (JCPDS no. 37-1493) [22]. The orthorhombic structure peaks are located at 24.02°, 33.08°, 35.61°, 40.93°, 49.37°, 54.00°, 57.57°, 69.60°, 71.78°, 75.35°, and 77.69°. Both La_2_O_3_ (JCPDS no. 73-2141) and La(OH)_3_ (JCPDS 13-1481) [23] phases are observed, and La_2_O_3_ and La(OH)_3_ exhibit hexagonal structures [24]. The peaks corresponding to La(OH)_3_ are 39.36° and 43.46°, while the peaks of La_2_O_3_ are 62.44° and 64.02°. In general, a higher intensity of XRD peaks indicates better crystallinity; thus, the lattice arrangement was highly ordered for the samples regardless of the Ni concentration.

The particle size increases as the Ni concentration increases due to the difference in the ionic radius of Fe^3+^ and Ni^2+^ [25]; the particle sizes shown in Figure 2 are listed in Appendix A. The specimen where x = 0.07 Ni has the largest particle size, approximately 7.82 ± 0.80 μm, because La_2_O_3_ and La(OH)_3_ appear in the matrix and enlarge the coverage area. Moreover, the morphologies of the sea urchin-like Au NPs decorated on the LFNO thin film are shown in Figure 3. The flower-like Au NPs are successfully adhered to the films for all LFNO samples with different Ni concentrations. The average size of the Au NPs is approximately 109.83 ± 8.48 nm.

### 3.2. Crystalline Analysis

Figure 4 and Figure 5 show the selected area diffractions (SAD) results and HR-TEM images of the raw LFNO powders, respectively. The SAD results, shown in Figure 4, are consistent with the XRD patterns. These raw LFNO powders exhibit high degrees of crystallinity, and the orthorhombic perovskite structures are dominant. As shown in the HR-TEM images in Figure 5, the d-spacing decreases when the sample is doped with a higher Ni concentration, which occurs because Ni atoms and NiO NPs induce lattice shrinkage [26]. Thus, the hydrogen molecules (H^+^) cannot be accommodated on the denser interlayer spacing, and hydrogen gas (H_2_) can be much more rapidly released. Then, the d-spacing is decreased with high Ni concentrations, and H^+^ or H_2_ is not incorporated into the lattices of the raw LFNO samples. Therefore, the H_2_ generation efficiency is improved.

### 3.3. Component Analysis

The EDX analysis results of the Au-coated LFNO are shown in Figure 6, wherein the elemental quantitative analyses are presented in sub-figures (c), (f), (i), (l), and (o). These results confirm that sea urchin-like Au NPs are adhered to each LFNO thin film.

Figure 7 shows the XPS patterns of all samples. Figure 7a,d,g,j,m are the O 1s state, and there are two peaks to show up for all O 1s state curves. One peak with intensity up to approximately 530.20 eV is corresponding to the oxygen vacancy (O_v_). Protons from the moist environment are exchanged with doped holes and oxygen vacancies formed by B-site cation doping [27], thus generating the more oxygen vacancies. Then, Ni is doped into the lattice of LaFeO_3_ also to induce the O_v_ enrichment. Another peak is the metal–oxygen (M-O) bond. The M-O bonds of Figure 7a,d,g,j,m indicate Fe-O bonds [28] and Ni-O bonds [29]. Thus, Fe_2_O_3_ and NiO are demonstrated to precipitate into the matrix. No other oxygen states are indicated. Based on Figure 7b,e,h,k,n, La 3d and Ni 2p orbits are overlapped. The peaks corresponding to the normal network structure of La^3+^ 3d_5/2_ and 3d_3/2_ are at approximately 838.3 and 847.7 eV, respectively. The La state in these results is consistent with the results reported in previous studies [30]. La_2_O_3_ is deposited, La-O bonds are decomposed, and oxygen atoms arrive at the interface to induce oxidation. Some peaks of La state disappear due to NiO formation. The La 3d_3/2_ orbit overlaps with the Ni 2p_3/2_ orbit. Only one peak of Ni^2+^ exists at approximately 870.3 eV [31], and no traces of other Ni states are observed. As seen in Figure 7c,f,i,l,o, one peak of Fe 2p_3/2_ exists at approximately 711.3 eV, and another peak of Fe 2p_1/2_ is at approximately 724.9 eV [32]. Fe is maintained at the +3 state, and no other valence sates are observed.

### 3.4. Absorbance Spectra of LFNO

The UV-vis spectra of raw and Au-coated LFNO are shown in Figure 8a,c, respectively. The energy gap (E_g_) and photonic extinction (PE) are determined from the UV-vis spectra, and these two energy levels are calculated with Equation (1) [33]. The diagram of (αhυ)^n^ vs. energy is shown in Figure 8b,d. In Equation (1), α is the absorption coefficient, hν is the photo energy, B is a constant, and n is defined the direct semiconductor (n = 2) or the indirect semiconductor (n = 1/2) [34]. Thereafter, a Tauc plot is used to measure E_g_ and PE. A value of n = 2 indicates that LFNO is a direct band gap material. Appendix A shows the parameters of the energy and absorbance wavelengths, obtained from Equation (1) and Figure 8b,d for different Ni concentrations. The following observations are noted: (1) E_g_ decreases as the Ni concentration increases because the NiO can be the active center of the infrared emission [35]; (2) PE tends to move to the visible region after the materials are decorated with sea urchin-like Au NPs; and (3) the maximum absorbance wavelength is augmented. In particular, the PE of Au-coated LaFe_0.93_Ni_0.07_O_3_ is only 1.99 eV, which can be attributed to the following reasons: (1) metal nanostructures scatter light strongly on the sharp tips of sea urchin-like Au NPs; therefore, the SPR are enhanced by several orders of magnitude compared with the incident light and tunes into the near-infrared region [36]; (2) a larger particle size induces the rough surface that can disperse the incident light; and (3) the intrinsic property of the LFNO sample contributes to the light absorbance. Moreover, after coating Au NPs, the PEs are also higher than 1.23 eV (H^+^/H_2_), meaning the photo–electro–chemical reaction is more toward HER. As seen in Figure 8e, the maximum absorbance wavelength of the prepared Au colloid solution is 535.2 nm. Therefore, the SPR band is at −2.32 eV (vs. vacuum).
(αhυ)^n^ = B(hυ − E_g_)(1)

### 3.5. Photo–Electro–Chemical Characteristics of Raw LFNO Powders

The valence band level, E_VB_, is calculated using Equation (2) [37], where hν is 21.1 eV, E^F^_B_ is the electron binding energy, and Φ_SA_ is the work function. Figure 9a–c show the UPS results for all LaFe_1−x_Ni_x_O_3_ for each Ni concentration. E^F^_B_ and Φ_SA_ are measured from the Tauc plot shown in Figure 9b,c, respectively. Finally, the E_VB_ is estimated using Equation (2) and Figure 9b,c. In addition, the conduction band energy, E_CB_, is evaluated.

Figure 9d shows the Mott–Schottky plots of the samples, where the slope crossover with the potential axis indicates the Fermi level. The carrier density is calculated using Equation (3) [38], where ξ is the dielectric constant of the film, ξ_0_ is the vacuum permittivity, e is the electric charge potential, T is the temperature in Kelvin, k is the Boltzmann constant, V is the applied voltage, V_FB_ is the flat-band potential, N_D_ is the donor density, and C_SC_ is the space charge capacitance. The carrier density increases as the Ni concentration increases, and greater carrier density means that electric and thermal conductivity are promoted. All raw LFNO powders are n-type semiconductors based on the positive slopes of Figure 9d.
E_VB_ = hυ − (E^F^_B_ − Φ_SA_)(2)
1/C^2^_SC_ = 2[V − V_FB_ − (kT/e)]/ξξ_0_eN_D_(3)

Figure 9e shows the C-V curves, and Figure 9f presents the LSV results. As the Ni concentration increases, the detection range of the redox reaction becomes broader, and LaFe_0.93_Ni_0.07_O_3_ demonstrates the broadest detection range. Additionally, the oxidation reaction is suppressed as the Ni concentration increases. For LSV, LaFe_0.93_Ni_0.07_O_3_ exhibits the largest HER potential. This result implies that the increased particle size with a high Ni concentration can contribute to a large number of active sites and macro-pores for oxygen transportation [39]. Appendix A shows the values of anodic (E_p_^a^), cathodic (E_p_^c^), and HER derived from the C-V curves and LSV analysis.

### 3.6. H_2_ Evolution and Activation Energy of the Au-Coated LFNO

Figure 10a–h show the I-V characteristics of the Au-coated LFNO in ethanol and 1-butanol at different temperatures. The H_2_ efficiency is calculated using Equation (4) [40], where η is the applied bias compensated solar-to-hydrogen efficiency, Ps is the energy of light irradiation, I_op_ is the photo-current density, V_op_ is the applied voltage, and the water electrolysis voltage is 1.23 V. From Figure 10 and Equation (4), the following phenomena are observed. First, the slopes and current densities of the samples tested using ethanol are larger than those of the samples tested using 1-butanol. Second, the H_2_ efficiencies of all specimens measured in ethanol are approximately 10 times larger than those measured in 1-butanol at any testing temperature. The reason for this is that the resistance of 1-butanol is higher than the resistance of ethanol, as 1-butanol has four carbon atoms in one molecule. The higher resistance results in a loss of conductivity (ohmic loss) and proton activity (kinetic loss) [41]. Equation (5) [42] is used to calculate the real production, where η is the H_2_ generation efficiency, P_s_ is the light power (1000 W/m^2^), A is the area (10^−4^ m^2^ for all samples), 237.1 × 10^3^ is the energy required for forming one mol of H_2_, and 2.0158 is the molecular weight of one mol of H_2_. The parameters listed in Appendix A represent the data obtained from Figure 10 and Equation (4). Tijare et al. found that LaFeO_3_, owing to the high visible-light activity, demonstrated the maximum H_2_ generation (3315 μmol g^−1^ h^−1^) [43]. Moreover, the H_2_ evolution rate of CdS-Au/MoS_2_ was up to 7010 μmol g^−1^ h^−1^ [19]. In this study, the real H_2_ production of all samples is a lot larger than 3315 μmol g^−1^ h^−1^, regardless of the temperature or solvent. However, in this study, the greatest amount of real H_2_ production is approximately 43,800 μmol g^−1^ h^−1^ in ethanol. Some representative perovskites (PVKs) for photocatalytic water splitting are listed in Table 1 [44,45,46,47,48,49,50,51,52,53,54,55,56,57,58,59].

The activation energies, E_a_, are obtained using the Richardson plot and Equation (6) [60], where I_0_ is the reverse saturation current in the I-V curves, T is the temperature in Kelvin, k is the Boltzmann constant, Φ_bo_ is the zero bias effective barrier height, q is the electronic charge, A is the working electrode area, and A* is the Richardson constant. Figure 10i,j show the Richardson plots for samples tested in ethanol and 1-butanol, respectively. From these results, the E_a_ of all Au-coated LFNO specimens measured in ethanol are smaller than the E_a_ measured in 1-butanol. This result implies that electrons are rapidly and easily transferred to H^+^ in ethanol, thereby generating more H_2_. For the definition of the kinetic loss, the sequential proton loss electron transfer mechanism includes two steps: (1) the deprotonation of the antioxidant, and (2) the transfer of an electron to the free radical. The compounds that can react these continuous processes result in the radical adduct formation, hydrogen transfer, and single electron transfer [61]. For 1-butanol, the low activation energy indicates that the pKa is also low, resulting in the slow proton exchange reaction. The values of the activation energies of each Au-coated specimen obtained from Equation (6) and Figure 10i,j are presented in Appendix A.
η = |1.23 − V_op_| × I_op_/P_s_ (P_s_:1000 [W/m^2^])(4)

Real production:[μmolg^−1^h^−1^] = η × P_s_ × A × 237.2 × 10^3^ [J/mol] × (2.0158 [mol/g] × 3600 [s]) × 10^6^(5)
I_0_ = AA*T^2^exp(−qΦ_b0_/kT)(6)

### 3.7. Energy Level Diagram and Scheme of H_2_ Generation

Figure 11a shows the energy level diagram of the raw LFNO samples with different Ni concentrations. The values of E_VB_, E_g_, E_CB_, E_FB_, and water splitting for the energy level vs. vacuum are shown. Appendix A indicates the parameters of the energy level diagram of raw LFNO (vs. vacuum) from Figure 11a. All the band gaps of the raw LFNO samples are greater than 1.23 eV (H^+^/H_2_). Thus, these materials are suitable to act as photocathodes.

Figure 11b shows the main reactions of raw LFNO samples under AM 1.5G, and the Fenton-like reaction happens. When the light is luminated on the samples in the solvent, the photogenerated charges, the electrons, and OH^•^ radicals appear. Then, the electrons transfer to Fe^3+^, and OH^•^ radicals move to Fe^2+^. The redox reactions are repeated and recycled between Fe^3+^ and Fe^2+^. The reduction reaction of Fe^3+^ to Fe^2+^ and the oxidization reaction of Fe^2+^ to Fe^3+^ are described by Equation (7) and Equation (8) [62]. The prime amounts of OH^•^ radicals are produced using Fe^3+^ under AM 1.5G, and OH^•^ radicals also simultaneously react with NiO. Equation (9) [63] indicates the oxygen gas generated under the visible light. Ni–Fe compounds are beneficial for the water dissociation into H^+^ and OH^−^ ions [64].
Fe^3+^ + H_2_O_2_ → Fe^2+^ + OH^•^ + H^+^(7)
Fe^2+^ + H_2_O_2_ → Fe^3+^ + OH^•^ + OH^−^(8)
O_2_ + H^+^ + e^−^ + hυ → 1/2 H_2_O_2_ + 1/2 O_2_(9)

Furthermore, Figure 11c shows the H_2_ evolution scheme of the Au-coated LaFe_0.93_Ni_0.07_O_3_ sample under AM 1.5G. Subsequently, Au supports the more optimized condition, which adds the extra active sites and allows the electrons to transfer rapidly for the hydrogen adsorption and the hydrogen gas formation. The hot electrons overcome the barrier at the interface between Au NPs and LaFe_0.93_Ni_0.97_O_3_ owing to SPR. The SPR energy is −2.32 eV. Therefore, more electrons are tunneled to the conduction band of LaFe_0.93_Ni_0.97_O_3_, leading to H_2_ generation. For Au-loaded semiconductors, the abilities of harvesting and absorbing light are promoted [65]. In this study, connecting the optical and electrocatalytic reactive interface between the LFNO semiconductor and the Au NP as a PEC cell can drastically enhance the H_2_ production efficiency in ethanol. This hybrid structure is satisfied with these conditions, the narrow band gaps, the rough surface, the contributions of SPR of Au NPs, and the coupling effect between LFNO-Au. Importantly, the energy in loss is decreaseddue to the plasmon-induced carriers of Au NPs [66], and there are the added active sites which arise from Au NPs. Thus, the photogenerated hot electrons provide high efficiency to generate the H_2_ reduction reaction.

## 4. Conclusions

In this work, double perovskite LaFe_1−x_Ni_x_O_3_ decorated with urchin-like Au NPs were synthesized. All raw LaFe_1−x_Ni_x_O_3_ powders exhibited high crystallinities, as indicated by the XRD and SAD patterns. In addition, La_2_O_3_ and La(OH)_3_ were found to co-exist. With the increasing Ni concentration, the following advantages were noted: (1) the generation of more active sites, (2) a lower band gap, (3) a higher carrier density, (4) a wider range of redox reaction, (5) HER potential toward a more positive value, and (6) smaller d-spacing. After coating with Au NPs, the PEs of all specimens reduced below 2.10 eV, and the absorbance wavelength occurred in the redshift due to the light scattering using Au NPs, the rough surface, and the intrinsic property of the samples. More light in the visible region was absorbed and harvested due to the hybrid structure. Moreover, larger specimen particle sizes induced more active sites. The average size of the sea urchin-like Au NPs was approximately 109.83 ± 8.48 nm. All the band gaps of the raw LFNO samples were higher than 1.23 eV (H^+^/H_2_). The H_2_ efficiency was 10 times higher in ethanol than that in 1-butanol because ohmic and kinetic losses resulted in the electrons or protons transferring on the surface of the electrode slowly in 1-butanol. That ethanol had lower activation energy resulted in the charges and protons transferring rapidly. The redox reactions were recycled among Fe^3+^, Fe^2+^, NiO, and hydroxyl. More photogenerated carriers were tunneled from Au NPs to the E_CB_ of the raw LFNO samples due to the SPR, the coupling effect, and the additionally active sites from Au NPs. This heterojunction structure was proven to develop the HER process.

## Figures and Tables

**Figure 1 nanomaterials-12-00622-f001:**
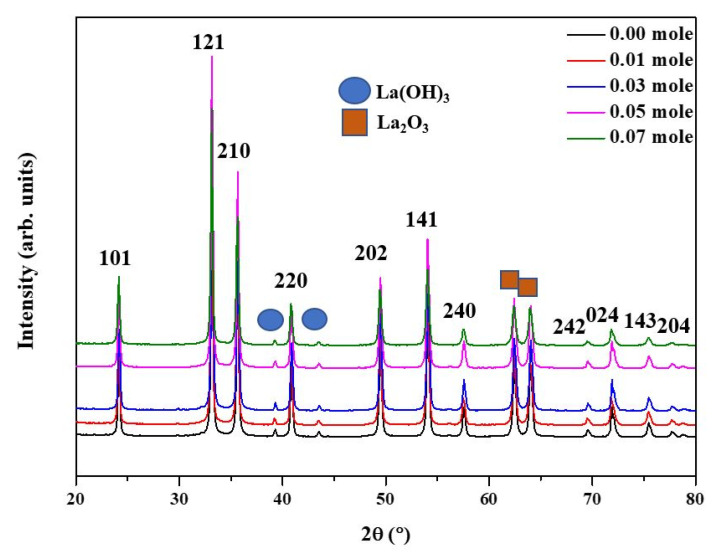
X-ray patterns of the raw LFNO samples with different Ni concentrations.

**Figure 2 nanomaterials-12-00622-f002:**
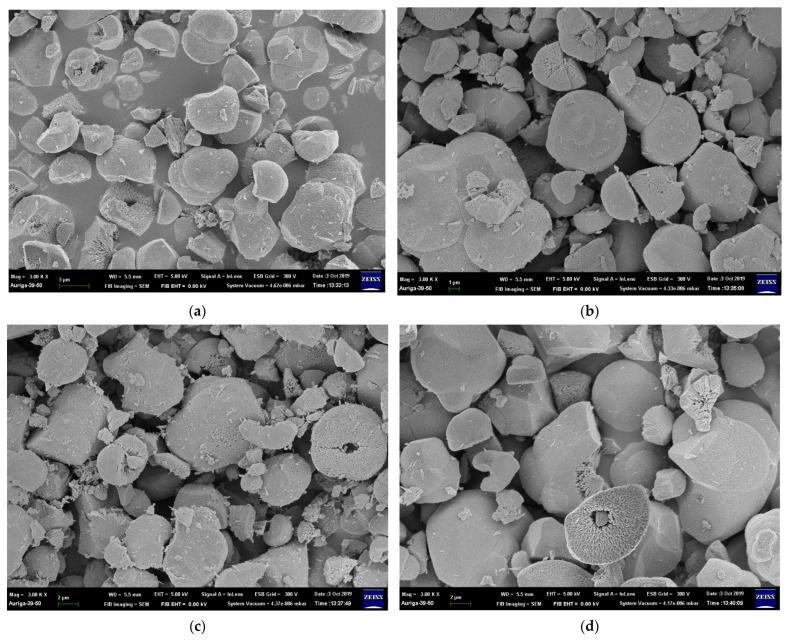
FE-SEM images of the raw LaFe_1−x_Ni_x_O_3_ in (**a**) x = 0.00 mol, (**b**) x = 0.01 mol, (**c**) x = 0.03 mol, (**d**) x = 0.05 mol, and (**e**) x = 0.07 mol.

**Figure 3 nanomaterials-12-00622-f003:**
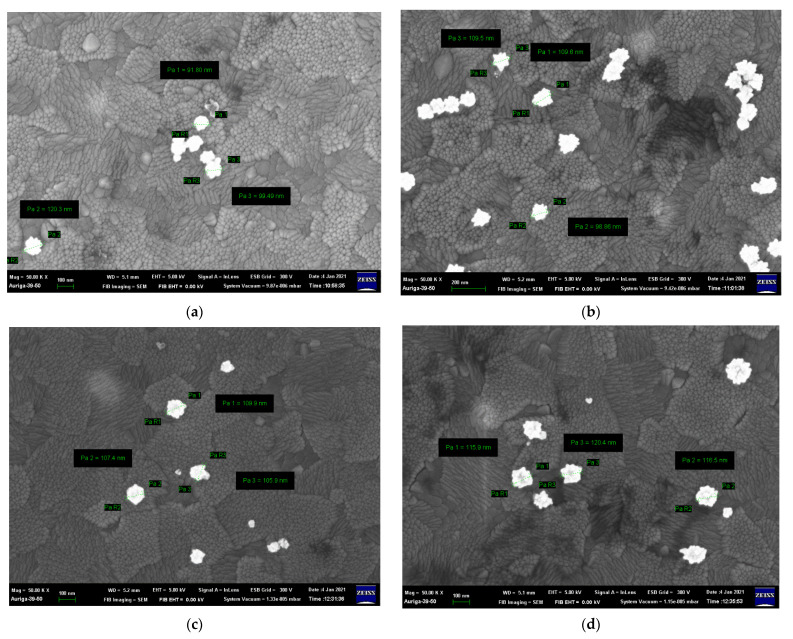
FE-SEM images of the Au-coated LaFe_1−x_Ni_x_O_3_ in (**a**) x = 0.00 mol, (**b**) x = 0.01 mol, (**c**) x = 0.03 mol, (**d**) x = 0.05 mol, and (**e**) x = 0.07 mol.

**Figure 4 nanomaterials-12-00622-f004:**
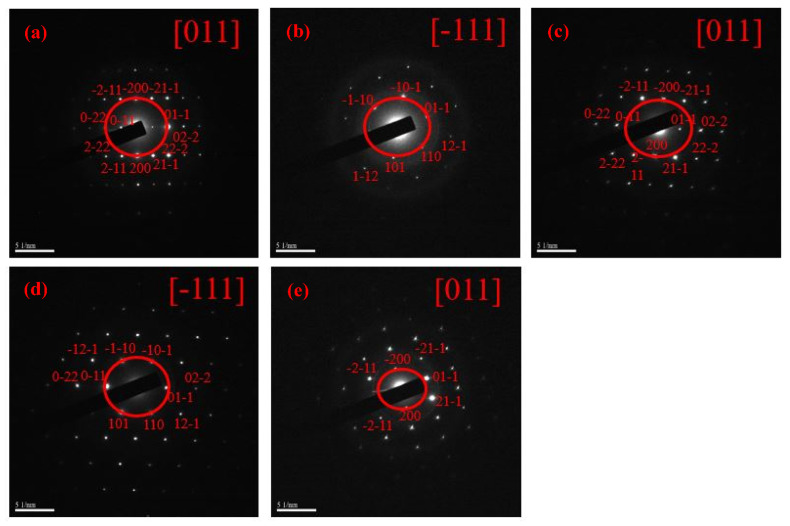
SAD of the raw LFNO samples in (**a**) x = 0.00 mol, (**b**) x = 0.01 mol, (**c**) x = 0.03 mol, (**d**) x = 0.05 mol, and (**e**) x = 0.07 mol.

**Figure 5 nanomaterials-12-00622-f005:**
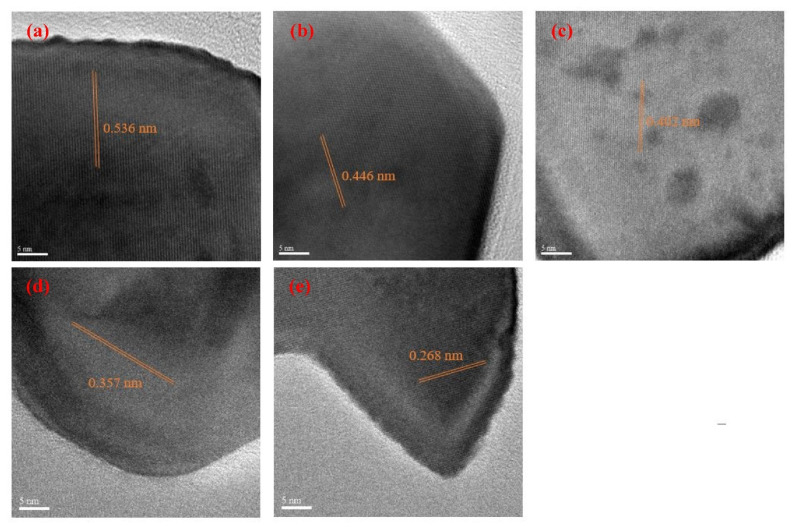
HR-TEM of the raw LFNO samples in (**a**) x = 0.00 mol, (**b**) x = 0.01 mol, (**c**) x = 0.03 mol, (**d**) x = 0.05 mol, and (**e**) x = 0.07 mol.

**Figure 6 nanomaterials-12-00622-f006:**
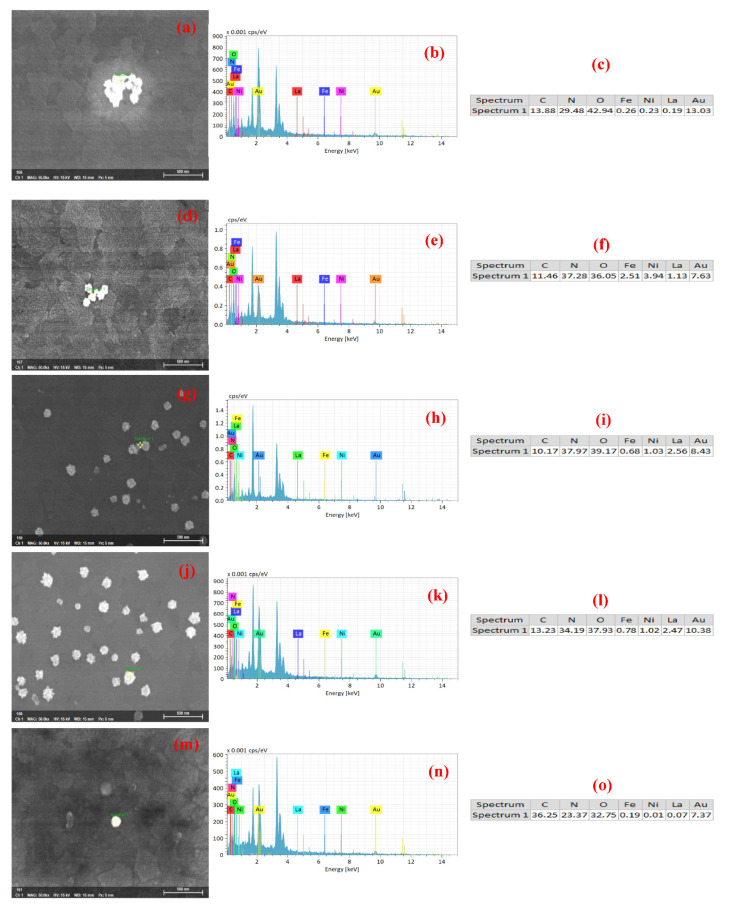
EDS analysis of the Au-coated LaFe_1.00_Ni_0.00_O_3_ sample in (**a**) original image, (**b**) energy spectrum, and (**c**) elemental composition (at %). EDS analysis of the Au-coated LaFe_0.99_Ni_0.01_O_3_ sample in (**d**) original image, (**e**) energy spectrum, and (**f**) elemental composition (at %). EDS analysis of the Au-coated LaFe_0.97_Ni_0.03_O_3_ sample in (**g**) original image, (**h**) energy spectrum, and (**i**) elemental composition (at %). EDS analysis of the Au-coated LaFe_0.95_Ni_0.05_O_3_ sample in (**j**) original image, (**k**) energy spectrum, and (**l**) elemental composition (at %). EDS analysis of the Au-coated LaFe_0.93_Ni_0.07_O_3_ sample in (**m**) original image, (**n**) energy spectrum, and (**o**) elemental composition (at %).

**Figure 7 nanomaterials-12-00622-f007:**
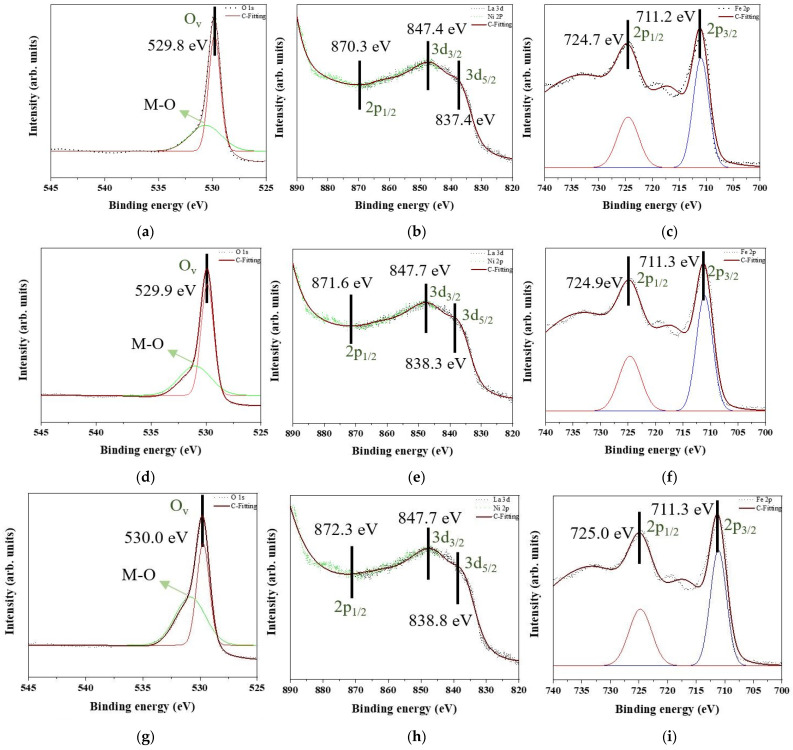
XPS patterns of the raw LaFe_1.00_Ni_0.00_O_3_ sample in (**a**) O 1s, (**b**) La 3d and Ni 2p, and (**c**) Fe 2p. XPS patterns of the raw LaFe_0.99_Ni_0.01_O_3_ sample in (**d**) O 1s, (**e**) La 3d and Ni 2p, and (**f**) Fe 2p. XPS patterns of the raw LaFe_0.97_Ni_0.03_O_3_ sample in (**g**) O 1s, (**h**) La 3d and Ni 2p, and (**i**) Fe 2p. XPS patterns of the raw LaFe_0.95_Ni_0.05_O_3_ sample in (**j**) O 1s, (**k**) La 3d and Ni 2p, and (**l**) Fe 2p. XPS patterns of the raw LaFe_0.93_Ni_0.07_O_3_ sample in (**m**) O 1s, (**n**) La 3d and Ni 2p, and (**o**) Fe 2p.

**Figure 8 nanomaterials-12-00622-f008:**
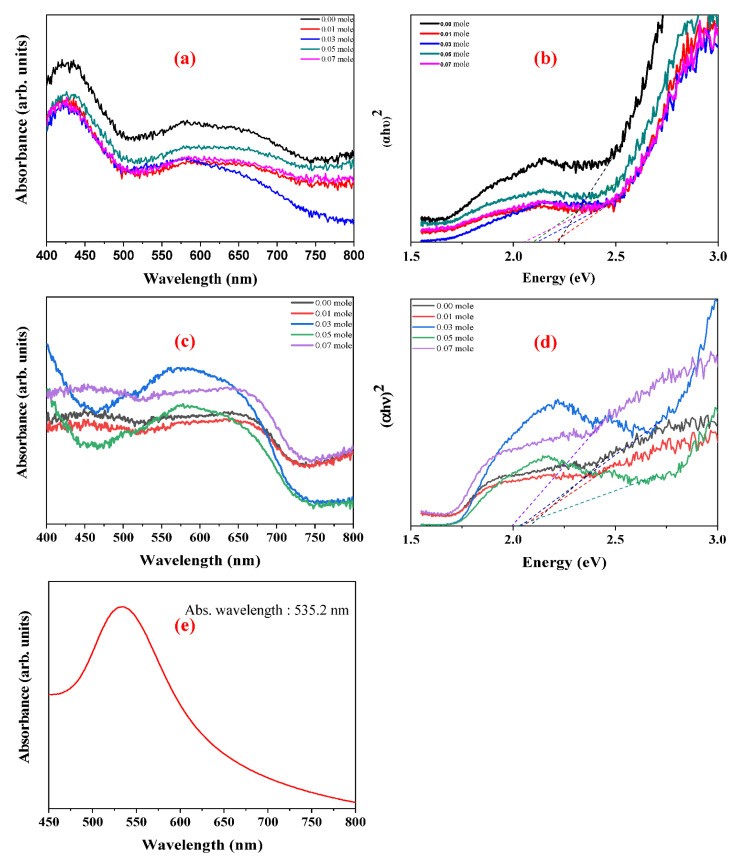
UV_−_vis spectra in (**a**) the raw LFNO samples, (**c**) the Au_−_coated LFNO samples, and (**e**) the Au colloid solution. (αhν)^2^ vs. energy in (**b**) the raw LFNO samples, and (**d**) the Au-coated LFNO samples. Energy was calculated from the Tauc plot.

**Figure 9 nanomaterials-12-00622-f009:**
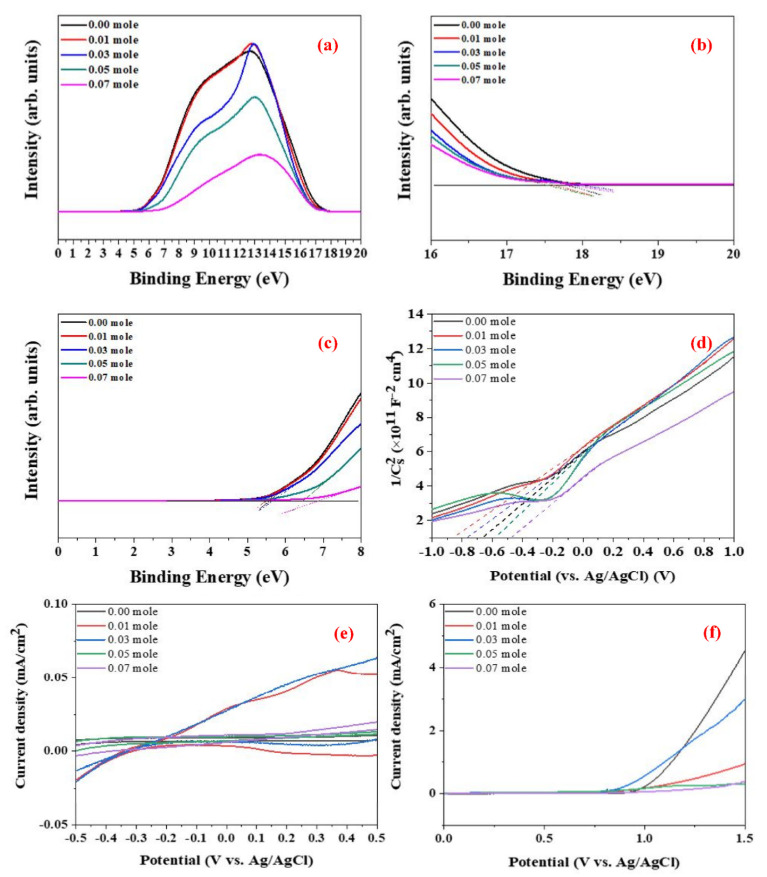
UPS spectra of the raw LFNO samples in (**a**) total, (**b**) electron binding energy, E^F^_B_, and (**c**) work function, Φ_SA_; these two parameters were calculated from the Tauc plots. (**d**) Electrochemical analysis in Mott_−_Schottky plots from which Fermi levels were calculated. (**e**) C_−_V curves of the raw LFNO samples in 0.1 M KOH, and (**f**) LSV of the raw LFNO samples in 0.1 M KOH solution under AM 1.5G.

**Figure 10 nanomaterials-12-00622-f010:**
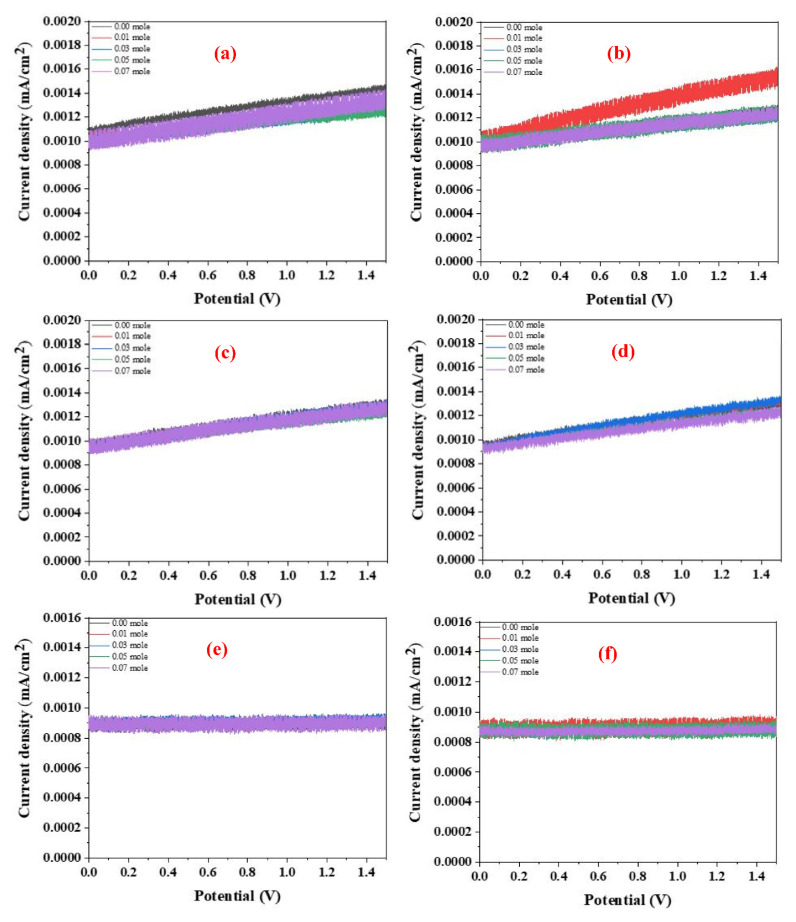
I_−_V curves of the Au_−_coated LFNO samples in ethanol with different temperatures under AM 1.5G in (**a**) at 0 °C, (**b**) at 10 °C, (**c**) at 20 °C, and (**d**) at 30 °C. I_−_V curves of the Au_−_coated LFNO samples in 1_−_butanol with different temperatures under AM 1.5G in (**e**) at 0 °C, (**f**) at 10 °C, (**g**) at 20 °C, and (**h**) at 30 °C. Richardson plots of the Au_−_coated LFNO samples were tested in (**i**) ethanol and (**j**) 1_−_butanol.

**Figure 11 nanomaterials-12-00622-f011:**
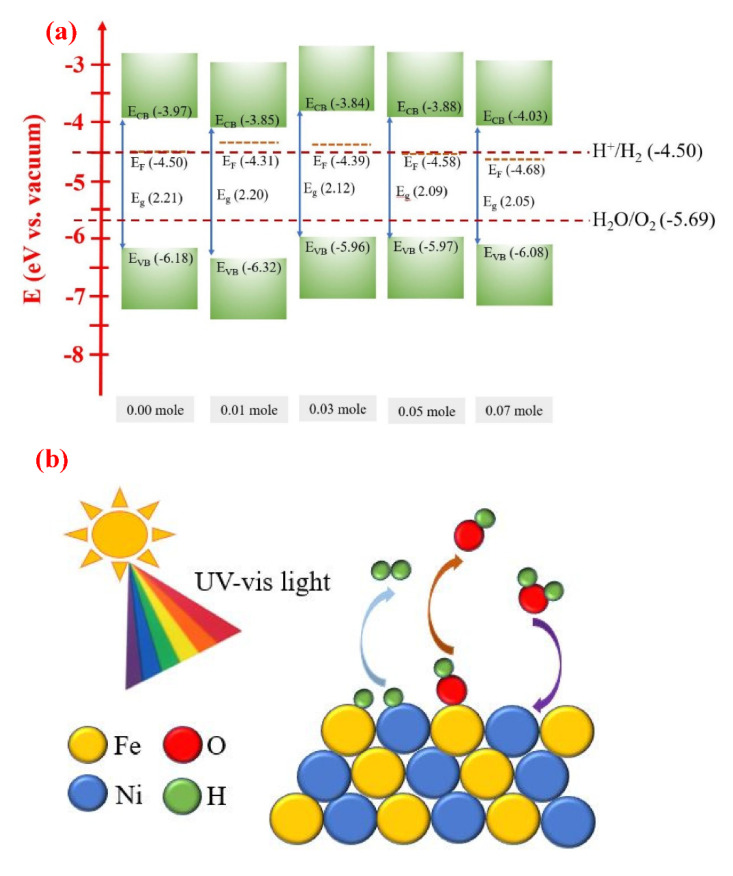
(**a**) Energy level diagram of the raw LFNO samples with different Ni mol concentrations, (**b**) the main reactions of photogenerated charges of the raw LFNO samples under AM 1.5G, and (**c**) H_2_ evolution scheme of the Au-coated LaFe_0.93_Ni_0.07_O_3_ sample under AM 1.5G.

**Table 1 nanomaterials-12-00622-t001:** The comparison of the photocatalytic HER performance of PVKs.

PVKs	Band Gap (eV)	Incident Light (nm)	Max. Real Production (μmol g^−1^ h^−1^)	Ref.
SrTiO_3_	2.97	≥360	3400.0	[44]
Na_1−x_K_x_TaO_3_	3.75	UV	11,000.0	[45]
LaNi_0.5_Cu_0.5_O_3_	2.47	<400	1166.0	[46]
LaKNaTaO_5_/LaTaON_2_	2.00	>420	≈110.0	[47]
K_2_LaTa_2_O_6_N	2.5	≥400	≈320.0	[48]
Ba_2_FeNbO_6_	2.29	>420	143.0	[49]
LaTiO_2_N	2.10	>420	3680	[50]
RhLaTa_2_O_6.77_N_0.15_	2.07	≥420	152.0	[51]
LiCuTa_3_O_9_	2.48	>420	0.70	[52]
LaMg_1/3_Ta_2/3_O_2_N	1.90–2.10	>300	5.00	[53]
Bi_4_NbO_8_Br	2.44	>420	83.30	[54]
Cs_2_AgBiBr_6_	1.65	>420	48.90	[55]
DMASnI_3_	1.30	>420	3.20	[56]
MAPbI_3_	≈1.51	≥420	758.9	[57]
MAPbBr_3−x_Ix	2.00	≥420	2604.8	[58]
CsPbBr_3−x_Ix	2.17	≥420	1120.0	[59]
Au-coated LaFe_1.00_Ni_0.00_O_3_	2.08	≥400	41,838.4	This work
Au-coated LaFe_0.99_Ni_0.01_O_3_	2.05	≥400	43,800.0	This work
Au-coated LaFe_0.97_Ni_0.03_O_3_	2.04	≥400	39,877.2	This work
Au-coated LaFe_0.95_Ni_0.05_O_3_	2.02	≥400	42,492.1	This work
Au-coated LaFe_0.93_Ni_0.07_O_3_	1.99	≥400	41,838.4	This work

## Data Availability

Image files must not be manipulated or adjusted in any way that could lead to misinterpretation of the information provided by the original image. Irregular manipulation includes (1) introduction, enhancement, moving, or removing features from the original image, (2) grouping of images that should obviously be presented separately (e.g., from different parts of the same gel, or from different gels), or (3) modifying the contrast, brightness or color balance to obscure, eliminate or enhance some information.

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
