# Peer review of "Double Perovskite LaFe1−xNixO3 Coated with Sea Urchin-like Gold Nanoparticles Using Electrophoresis as the Photoelectrochemical Electrode to Enhance H2 Production via Surface Plasmon Resonance Effect"

_nanomaterials, 2022, doi:10.3390/nano12040622_

Round 1
Reviewer 1 Report
The authors produced a series of double perovskites LaFe(1-x)Ni(x)O3 particles decorated with sea-urchin-like Au nanoparticles. The nanoparticles were systematically and comprehensively studied. The authors found that the H2 efficiency corresponding to the increase of the active sites to develop HER process. The result may be new for NCO and is supported by the extensive experiments. Therefore, the reviewer recommends publication in “nanomaterials” after following technical improvements for figures.
*Low figure resolution: Figs. 1, 2, 3,
*Lack of scale bar: Fig. 1
*Too small figures and characters inside. Some of characters can not be read because of blurred and/or too small.: Figs. 1, 2, 3, 5, 6, 7
Reviewer 2 Report
In the present report, the authors constructed a series of double perovskites LaFe1-xNixO3 was synthesized. Further, these LaFe1-xNixO particles were then decorated with sea-urchin-like Au nanoparticles using electrophoresis. Moreover, the hot electrons from the Au nanoparticles were injected via the SPRs effect, the coupling effect between LaFe1-xNixO3 and Au nanoparticles, and the more active sites from Au nanoparticles into the conduct band of the semiconductor, improving the hydrogen efficiency. Nevertheless, the H2 efficiency of the Au-coated LaFe1-xNixO3 measured in ethanol was approximately ten times larger than the that of Au-coated LaFe1-xNixO3 measured in 1-butanol at any testing temperature. This manuscript provided some valuable information and the content is very significant in this field. However, I recommended a major revision of the article from its present form before it can be published in nanomaterials. Some specific comments are as follows:
- Introduction line 2 the word “constitutions” can be replaced by “composition”
- Introduction – “conduct band” can be replaced by “conduction band”
- What is mean by “unas-sisted water-splitting reaction”. Please explain
- Introduction writing style and connectivity needs to be improved and more polished
- In the introduction section, the authors should expound the research significance of the present work. The authors should explain the novelty of the present report?
- A comparison on why Double perovskites are admired as photocatalysts than the traditional materials?
- Separate the XRD picture. Its hard to read and observe the figure, peaks and labels inside it.
- What is the reason behind choosing electrophoresis for Au depsotion? Because the particle size is too high, almost in micron range.
- Is it possible to reduce the particle size of LaFe1-xNixO3, by other synthesis methods?
- Figure 3 quality needs to be improved definitely. Its hard to see the data. Same like Figure, 7. The legends inside the figure are almost invisible to the reader.
- A lot of other literature is available on Perovskite-based catalysts. Please input a tabular form of comparison with the existing literature.
- Suggested to cite following papers, 10.1016/j.jechem.2020.08.057, 10.1002/anie.201900292, 10.3390/en15010272
- In the current state, there are more typographical errors and the language should be improved. Therefore, the authors are advised to recheck the whole manuscript for improving the language and structure carefully.
Round 2
Reviewer 2 Report
The authors have addressed all of the reviewer's comments satisfactorily and therefore the manuscript can now be accepted for publication in the present form.